# Imidazole: Synthesis, Functionalization and Physicochemical Properties of a Privileged Structure in Medicinal Chemistry

**DOI:** 10.3390/molecules28020838

**Published:** 2023-01-13

**Authors:** Heber Victor Tolomeu, Carlos Alberto Manssour Fraga

**Affiliations:** 1Laboratório de Avaliação e Síntese de Substâncias Bioativas (LASSBio), Instituto de Ciências Biomédicas, Universidade Federal do Rio de Janeiro, Rio de Janeiro 21941-909, Brazil; 2Programa de Pós-Graduação em Farmacologia e Química Medicinal, Instituto de Ciências Biomédicas, Universidade Federal do Rio de Janeiro, Rio de Janeiro 21941-909, Brazil

**Keywords:** imidazole, synthesis, medicinal chemistry, drug discovery

## Abstract

Imidazole was first synthesized by Heinrich Debus in 1858 and was obtained by the reaction of glyoxal and formaldehyde in ammonia, initially called glyoxaline. The current literature provides much information about the synthesis, functionalization, physicochemical characteristics and biological role of imidazole. Imidazole is a structure that, despite being small, has a unique chemical complexity. It is a nucleus that is very practical and versatile in its construction/functionalization and can be considered a rich source of chemical diversity. Imidazole acts in extremely important processes for the maintenance of living organisms, such as catalysis in enzymatic processes. Imidazole-based compounds with antibacterial, anti-inflammatory, antidiabetic, antiparasitic, antituberculosis, antifungal, antioxidant, antitumor, antimalarial, anticancer, antidepressant and many others make up the therapeutic arsenal and new bioactive compounds proposed in the most diverse works. The interest and importance of imidazole-containing analogs in the field of medicinal chemistry is remarkable, and the understanding from the development of the first blockbuster drug cimetidine explores all the chemical and biological concepts of imidazole in the context of research and development of new drugs.

## 1. The Chemistry of Imidazole

Imidazole **1** (Figure 1) was first synthesized by Heinrich Debus in 1858, but since the 1840s, several imidazole derivatives have been discovered. Its synthesis started from the use of glyoxal **2** and formaldehyde **3** in ammonia, producing imidazole **1** as a final product, which was initially called glyoxaline (Figure 2) [1,2]. This synthesis, despite producing relatively low yields, is still used to create *C*-substituted imidazoles.

Among the nitrogen-based heterocyclic compounds, imidazole **1** plays an important role in humans. It is included in chemical sciences, biological sciences and materials science, and used as a catalyst in compound synthesis and the process of developing new drugs (Figure 3) [3,4,5,6].

With a 5-membered planar ring, imidazole **1** exhibits solubility in water and other polar solvents. Two equivalent tautomeric forms are observed because the hydrogen atom can be located on either of the two nitrogen atoms [2]. Imidazole **1** is a highly polar compound, as seen by a calculated dipole of 3.61D, and is completely soluble in water [2,7]. Imidazole **1** is classified as an amphoteric compound, acting as both an acid and a base. The compound is classified as aromatic due to the presence of a sextet of π electrons, consisting of a pair of nonbonding electrons from the nitrogen *N*-1 atom and one from each of the four remaining ring atoms [2,7].

Imidazole **1** can form stable crystalline salts with strong acids through the protonation of the *sp*^2^ nitrogen (*N*-3), known as imidazolium salts. Imidazole **1** has a pKaH of 7.1 (Figure 4), acting as a strong base [8]. The basicity of imidazole is above that of pyridine **4** (pKaH of 5.2) due to the amidine-like **5** resonance, which allows both nitrogens to participate equally in charge accommodation [7,8,9]. Comparatively, the basicity of imidazole **1** contrasts with the basicity of pyrrole **6** (pKaH of 0.4), which is an extremely weak base, because when pyrrole **6** is protonated, there is a loss of aromaticity which is built with the participation of the nonbonding electron pair of *N*-1 nitrogen [7].

Imidazole **1** is a good donor and acceptor of hydrogen bond interactions; the *sp*^2^ nitrogen (*N*-3) accepts a hydrogen interaction, while *N*-1 nitrogen, being relatively acidic, donates its hydrogen to an interaction (Figure 5A) [8]. This property is fundamental for the mode of action of several enzymes that use the imidazole ring, such as the histidine **7** amino acid residue, one of the 20 amino acids found in proteins [8]. These important interactions for living organisms are present not only in macromolecules but also in bioactive small molecules.

A recent study demonstrating this important effect was carried out by Movellan and collaborators [10], where it was possible to analyze the hydrogen residues in histidine residues **7** in the M2 tetramer of influenza A, important for the process of endocytosis and maintenance of the virus life cycle be [1,2,3,4,11,12,13,14,15]. Using the M2 conduction domain construct in lipid bilayers, that the imidazole ring is hydrogen bonded even at a pH of 7.8 in the neutral charge state (Figure 5B). An intermolecular ^2h^*J*_NN_ hydrogen bond of 8.9 ± 0.3 Hz was observed between H37 Nε and Nδ. However, this ^2h^*J*_NN_ interaction could not be detected in the sample connected to the drug rimantadine (Rmt), with consequent modification in the proton chemical shifts value of 3 ppm for histidine residues [10].

Histamine **8** is an example of a small molecule with different actions in living organisms. It is biosynthesized from histidine **7** itself by the action of the enzyme histidine decarboxylase (Figure 5B) [16].

The presence of the *N*-1 nitrogen in the imidazole **1** structure makes it tautomer, which becomes evident in non-symmetrically substituted compounds, such as methyl imidazole **9** (Figure 6) [9]. This curious feature of imidazole chemistry means that simply writing “4-methylimidazole” would be incorrect, considering the rapid tautomeric equilibrium with the 5-methylimidazole structure [9,17].

Despite the existence of the tautomeric effect in imidazole **1**, the ratio in terms of proportion of these tautomers varies according to the substituent added to the ring. The tautomer ratio observed in the 4(5)-nitroimidazole **10** derivative is approximately 400:1 for 4-nitroimidazole [9]. Reports in the literature suggest a relationship between tautomers 1.4 and 1.5 in the following proportion: log ([1.4]/[1.5]) = 4 × σ_m_ [17,18]. However, for any substituent, whether electron withdrawing (EWG) or electron donor (EDG), the 1,4 tautomer predominates since the meta substituent constant (σ_m_) is governed by inductive effects (σ_m_ = 0.71, 0.37 and 0.10 for -NO_2_, -Br and -OCH_3_, respectively) [17]. The 4(5)-nitroimidazole **10** derivative has a pKa of 9.30 (different from the pKa of its imidazole **1** precursor, pKa = 14.5) with a predominance of tautomer 1.4 (Figure 7) [19,20]. It is possible to notice these effects in a quick energy calculation for a geometric equilibrium in water using the PM3 semiempirical method, using the Spartan 18 v1.2.0 program (Wavefunction, Inc & Q-Chem, Irvine, California, U.S.A). There is a significant difference in the energy obtained for tautomer 1.4 (15.21 kJ/mol) in comparison with that of corresponding tautomer 1.5 (25.36 kJ/mol).

Imidazole **1** is an electron-rich heterocycle, as mentioned above. Electrophilic substitution normally occurs at the *C*-4 or *C*-5 position; however, as previously mentioned, the predominance of a given tautomer is relative and follows the factors already discussed, while nucleophilic substitution usually occurs at *C*-2 [7]. Calculations in Spartan 18 v1.2.0 help us obtain a better view of these data. Figure 8 shows the electrostatic map with the values of natural charge for imidazole **1** (−0.290, −0.013, −0.217, −0.194 and −0.092, for *N*-1, *C*-2, *N*-3, *C*-4 and *C*-5 respectively), demonstrating the electronic density for each atom of the imidazole ring, suggesting the electronegativity (χ) of the structure, as well as the dipole moment of imidazole **1**, which starts from *N*-1nitrogen toward *sp*^2^ nitrogen. With these results, we can consider the information taken from the literature, where *C*-4 and *C*-5 have higher electron densities, making them susceptible to electrophilic reactions, and *C*-2 has a lower density, making it susceptible to nucleophilic reactions.

Although imidazole **1** and pyrrole **6** are π-excessive heterocycles (π donors), imidazoles do not establish η^5^(π)-complexes (η being the chemical hardness) with transition metals. Imidazole **1** behaves as a π-deficient ligand (π-acceptor) similar to pyridine **4**, and its *sp*^2^ nitrogen atoms mainly form η^1^(σ,N_py_) complexes (Figure 9), as does pyridine **4** [7].

Imidazole **1** has a very exciting physicochemical complexity. This makes it a target nucleus for the most diverse applications, which employ synthetic methodologies for its obtention and functionalization.

## 2. Synthesis and Functionalization of Imidazole

The synthesis of substituted imidazole **1** using heterogeneous catalysis has been widely exploited. These functionalized structures are useful building blocks for the synthesis of molecules of biological and pharmaceutical interest.

### 2.1. Mono-Substituted Derivatives

One-pot reactions using iodobenzene **11** and imidazole **1** in the presence of K_3_PO_4_ as the base, CuI as the catalyst and DMF as the solvent at 35–40 °C for 40 h, give the corresponding *N*-arylimidazoles **12** in quantitative yields (Figure 10) [21].

Exploring the bifunctionalization of 1,2-disubstituted acetylenes **13** by ruthenium carbonyl to form *cis*-enediol diacetates **14**, followed by reaction with ammonium carbonate as a source of nitrogen and methanol for the *C*-2 carbon, permitted us to obtain monosubstituted imidazoles **15,** as shown below (Figure 11). In reactions where (R) were aromatic rings, both substituents, withdrawers (EWG) and donors (EDG) were tolerated under the applied conditions [22].

### 2.2. Disubstituted Derivatives

The work below shows the development of an efficient methodology for the synthesis of novel 2-aryl-4-benzoyl-imidazoles **16** by structural modification of 2-aryl-imidazole-4-carboxylic amide (AICA) **17** and 4-substituted methoxylbenzoyl-aryl-thiazoles (SMART) **18**, presenting antiproliferative activity (Figure 12) [23].

Disubstituted imidazoles **24** could also be synthesized by cyclization of α-keto-aldehydes **25** obtained from the oxidation of aryl methyl-ketones **26** with selenium dioxide (SeO_2_) after treatment with ammonium acetate and ethanol. The synthesized derivatives were used as fluorogenic sensors for detection, selectivity and sensitivity to Fe^3+^ ions (Figure 13) [24].

Another synthetic strategy used to construct disubstituted imidazole derivatives **27** from methyl ketones **28** consisted of exploiting its metal-free acid-catalyzed oxidation and coupling with aldehydes **29** and **30** in the presence of ammonium acetate (Figure 14) [25].

Moreover, disubstituted imidazoles presenting a carbomethoxy group **31** at *C*-4 could be obtained from the coupling of functionalized amidoximes **32** and methyl propiolate **33** in the presence of a catalytic amount of 1,4-diazabicyclo[2.2]octane (DABCO) under microwave irradiation (Figure 15) [26].

Amido-nitrile **34** cyclization mediated by functionalized boronic acids is also able to produce 2,4-disubstituted imidazoles **35**. It was possible to explore a considerable diversity of substituents, considering the reaction conditions reported below (Figure 16) [27].

### 2.3. Trisubstituted Derivatives

On the other hand, 2,4,5-trisubstituted imidazoles **36** could be obtained by using 2,3-dioxo-3-substituted propanoates **37** as precursors after condensation using ammonium acetate and various aromatic aldehydes **38** in EtOH and AcOH as catalysts at room temperature (Figure 17) [28,29].

2-Aryl-4,5-dicarbonitrile imidazole derivatives **39** could be obtained from the coupling of substituted aromatic aldehydes **40** and 2,3-diaminomaleonitriles **41** in the presence of a mixture of cerium (IV) ammonium nitrate/nitric acid (CAN: NA|0.05: 0.4 eq.) at 120 °C for less than 1 h without using solvents (Figure 18) [30].

The efficient combination of α-aminoketones **42** with formamide **43** in THF at 180 °C for 8 h is also able to provide 1,4,5-trisubstituted imidazoles **44** [31] (Figure 19).

Using the Van Leusen method, it was possible to synthesize 1,4,5-trisubstituted imidazoles containing a trifluoromethyl group **45**, exploiting the coupling of *N*-aryltrifluoroacetimidoyl chloride **46** and tosyl-methylisocyanate (TosMIC) **47** using sodium hydride as a base in dry THF at room temperature (Figure 20) [32].

Methodologies reported in the literature show the use of substituted 1,2-diphenylethane-1,2-dione (benzyl) **48**, substituted aldehydes **49** and ammonium acetate under various conditions, with the aim of optimizing the construction of 2,4,5-trisubstituted imidazole **50** with great structural diversity [33,34,35,36,37,38,39,40]. The reactions are simple and fast, as illustrated in the example below, which uses the system with EtOH and fluorinated graphene oxide (A-MFGO) as a catalyst at room temperature (Figure 21).

More complex heterocycles presenting the imidazole ring in their structure are described in the literature, such as benzo[*d*]imidazo[2,1-*b*]thiazoles **51**, which could be synthesized by the condensation of aromatic ketones **52** and 5-(biphenyl-4-yl)-1,3,4-thiadiazol-2-amine **53** in the presence of *N*-bromosuccinimide **54**, PEG-400 and water as solvent under microwave irradiation at 85 °C in quantitative yields after a few minutes of reaction (Figure 22) [41].

### 2.4. Tetrasubstituted Derivatives

Using various aldehydes **55**, benzyl **56**, ammonium acetate and prop-2-ynylamine **57** in the presence of CuFe_2_O_4_NPs as a catalyst in H_2_O:EtOH under reflux for approximately 50 min, it was possible to obtain several tetrasubstituted imidazole derivatives **58** in a multicomponent synthesis. It was possible to reuse the catalyst for six reactions without losing its efficiency (Figure 23) [42].

In the presence of SO_4_^2−^/Y_2_O_3_ as a catalyst, the multicomponent condensation of benzyl **56**, aminoethylpiperazine **59**, various aldehydes **60** and ammonium acetate in ethanol at 80 °C for 10 h was carried out to form tetrasubstituted 1,2,4,5-imidazole derivative **61**. The catalyst was reused up to five times with no significant loss in catalytic efficiency (Figure 24) [43].

Alternatively, the synthesis of 1,2,4,5-tetrasubstituted imidazole derivatives **62** could be achieved through the condensation of benzyl **56**, aldehydes **63** and anilines **64** in the presence of ammonium acetate under the solvent-free catalysis of Fe_3_O_4_@SiO_2_/bipyridinium nanocomposite (Fe_3_O_4_@SiO_2_/BNC) (Figure 25). The catalyst was reused until the fifth reaction without much change in catalytic efficiency. Methodologies using other catalysts and even solvents have also been reported [44,45,46,47].

Another methodology used for obtaining hybrid imidazole derivatives was starting with hippuric acid **65** and 2-chloroquinoline-3-carbaldehyde **66** in acetic anhydride, and 4-((2-chloroquinolin-3-yl)methylene)-2-phenyloxazol-5(4*H*)-ones **67** could be obtained through Perkin condensation in the presence of anhydrous sodium acetate under microwave irradiation. Subsequently, the previously obtained derivatives were condensed with *N*-aminoarylcarboxamides **68** in pyridine under reflux to furnish the final desired azaheterocyclic acylhydrazides **69**. For some derivatives of this series, antimicrobial properties were observed (Figure 26) [48].

Since its synthesis in 1858, the imidazole **1** ring has been exploited in different contexts, whether chemical or biological. The examples presented herein illustrate the more recent ways of obtaining this azaheterocyclic system through the use of a range of methodologies and chemical reagents, providing great chemical diversity.

## 3. Imidazole as a Privileged Structure in Medicinal Chemistry

As already mentioned, imidazole **1**, in biological systems, is found in the form of the amino acid histidine **7**, presenting an important role in the catalysis promoted by enzymatic systems [8]. Furthermore, the neurotransmitter histamine **8** induces immunological processes [8,16,49,50,51] and composes the structures of the guanine **70** and adenosine **71** bases of nucleic acids (Figure 27) [52].

### 3.1. The Catalytic Potential of Imidazole in Biological Systems

The breakage of the P-O and C-O bonds are biological events of extreme importance and are estimated to be on the order of 5 to 13 million years [53] in the absence of enzymes and can reach the order of billions of years for DNA [54]. Imidazole **1** stands out for constituting numerous enzymatic active sites in the form of the histidine **7** amino acid residue and acting in catalytic processes, accelerating such unfavorable reactions [55].

The imidazole **1** group in biological systems generally acts in acid–base and nucleophilic catalysis (Figure 28). As an acid catalyst, protonated imidazole **1** acts as an acid facilitating the exit of the RO- group through hydrogen transfer [56]. In the neutral form, imidazole **1** can act as a nucleophilic catalyst by attacking the electrophilic center, leading to a phosphorylated or acylated intermediate, which is consecutively hydrolyzed, regenerating the imidazole **1** group [57]. Therefore, imidazole **1** catalyzes the cleavage of the X-O bond (for X = C or P). Finally, imidazole **1** can also act as a basic catalyst, assisting the attack of a nucleophile (Nü-H) on the electrophilic center of a substrate, abstracting a proton and thus increasing its nucleophilicity [58,59,60,61].

The amino acid histidine **7** plays a fundamental role in several enzymatic active sites, including ribonucleases, phosphotriesterases, kinases, chymotrypsins and histone deacetylases [62,63,64,65]. As illustrated in Figure 29, histidine **7** residues (H573/H574) participate in the catalytic process of deacetylation of the lysine residue by histone deacetylase 6 (HDAC6). We can also see the residue (H614) acting on the triad of amino acids responsible for sustaining the zinc atom (through the interaction η^1^(σ,N_py_)) present in the catalytic site of the enzyme [7,65].

It is interesting to analyze in more detail the kinetic profiles considering the pH for an acid, basic, bifunctional acid–base and nucleophilic catalysis for the cases of deacylation reactions [66,67], but which similarly follow the same profiles in dephosphorylation [68,69]. In some pH ranges (considered pH 5-9), imidazole **1** has pronounced activity, with prevalence at pH values (considering pH 8-10) above the pKaH, for the neutral imidazole **1** species. This action makes it a basic and nucleophilic catalyst with similar kinetic profiles for both cases, where the rate constant is directly proportional to the increase in pH, capable of presenting a level where the amount of neutral and reactive species remains constant (pH ~9) (Figure 29) [49]. At the plateau, it is common to observe that the rate constant in nucleophilic catalysis is higher than that in basic catalysis, considering that nucleophilic processes are faster [55]. For pH values below the pKaH, where protonated imidazole **1** predominates, the catalysis is preferably acidic. As with basic and nucleophilic catalysis, acid catalysis has the same effect at the plateau (pH ~5) (Figure 29) [55].

Considering the above remarks, we can conclude that histidine **7** residues present in the active site of HDAC6 participate in catalysis by a combined acid/base mechanism, as this hydrolyses assists with the aid of the tyrosine residue (Y745) and the metal itself (Zn^2+^), which, when complexed with the carbonyl oxygen, further favors the reaction shown, considering the low reactivity of amide carbonyls (Figure 29).

### 3.2. Imidazole as a Building Block in the Structure of Bioactive Molecules and Drugs

Imidazole **1** is present in several chemical structures of pharmaceutical interest because its particular chemical properties could favor molecular recognition by different targets. Examples of the presence of imidazole **1** in the structure of bioactive substances include antibacterial [70,71,72,73], anti-inflammatory [74,75,76], antidiabetic [77], antiparasitic [78], antituberculosis [79], antifungal [80,81,82], antioxidant [83], antitumor [84,85,86], antimalarial [87,88], anticancer [89,90,91], antidepressant [92] and many other compounds (Figure 30).

Moreover, imidazole **1** is also present in several natural compounds with biological activity. As an example, pilocarpine **72** is used for the treatment of xerostomia and glaucoma [93], topsentin **73** shows anticancer activity [94] and isonaamine A **74** also shows anticancer activity through its action as an inhibitor of the epidermal growth factor receptor (EGFR) [95] (Figure 31).

To date, a great diversity of imidazole-containing compounds with biological activity is known, whether of natural or synthetic origin. Among the compounds with imidazole-containing activity, azomycin **75** can be considered the one with the simplest structural complexity of all. The antibiotic azomycin was first isolated by Maeda in 1953 from a strain similar to *Mesenteric nocardia* [96]. In addition to azomycin **75**, we can illustrate the structure of several more complex compounds that are even well known in current pharmacotherapy containing the imidazole nucleus, such as dacarbazine **76** [97], nafimidone **77** [98], flumizole **78** [99], cimetidine **79** [100], losartan **80** [101] and ketoconazole **81** [102] (Figure 32).

Imidazole **1** proves to be a very versatile structure for medicinal chemistry, not only for its ability to act directly as a pharmacophoric group but also for being able to act as a “guide” to other groups, directing them and favoring correct auxophoric/pharmacophoric interactions, allowing the exploration of a large number of possible substitutions. Imidazole **1** is considered a bioisostere of a carboxamide unit. Thus, it can be interpreted as a peptide backbone unit isostere [103]. Depending on the substituents and their substitution pattern, small mimetic oligopeptides with *trans* and *cis* conformations can be evidenced (Figure 33) [103,104]. The isosteric exchange of amides for imidazole can be considered a good strategy to overcome problems resulting from metabolic instability promoted by amidases [105].

In a more recent work by Heppner and collaborators [106], the importance of the imidazole nucleus in compounds containing biological activity was highlighted. The aim of the study was the modulation of the mutated epidermal growth factor receptor (EGFR) target, in the context of non-small-cell lung cancer, which presents acquired drug resistance. Compounds with nanomolar potency against EGFR (L858R/T790M/C797S) were obtained in a reversible binding mechanism [106].

Analysis of the X-ray crystallographic results shows how the imidazole nucleus acts as a hydrogen bond acceptor for the catalytic residue of lysine (K745) in the “αC-helix out” inactive state of EGFR. Furthermore, selective *N*-methylation on the imidazole nucleus at the hydrogen bond acceptor position drastically reduces the potency, confirming the importance of the interaction of (K745) with the imidazole nucleus for modulating the EGFR variant (C797S) [106]. Additionally, it was observed that there is an intramolecular hydrogen interaction between the *N*-1 and the phenylacrylamide group **82** (Figure 34) [107]. The covalent bond with (C797) does not significantly change the mode of inhibitory interaction compared to the reversible compounds, indicating that the interaction of imidazole with (K757) is conserved in both covalent and non-covalent modes.

Analyzing the activity values for **82**, **83** and **84** (Figure 34), it was concluded that *N*-1 methylation **83** impacts activity against EGFR (L858R/T790M), indicating that the intramolecular interaction between imidazole and the phenylacrylamide group is required for the inhibitory mode of interaction. For *N*-3 methylation **84**, the nitrogen involved in the interaction with (K745) in the inactive form “αC-helix out” of EGFR, does not significantly alter the inhibition activity towards EGFR (L858R/T790M), showing that the potency of **82** does not depend on the interaction with the residue (K745) but more on the covalent bond with (C797).

Finally, it was observed that, for EGFR (L858R/T790M/C797S), the presence of both unsubstituted nitrogens in the imidazole nucleus are extremely important for activity against this mutated form. Confirmed through the data already presented by **83** and **84**, as well as the data for **85** and **86**, showing excellent IC_50_ values with this interaction pattern, and acting as reversible inhibitors. 

This study shows us as an interesting practical example how imidazole may be able to act as an important auxophoric group and act directing other auxophoric and pharmacophoric groups.

Another example of imidazole application was described by Lee et al. [108], where the activity-based sensing (ABS) strategy for detecting copper in living cells was presented, which preserves spatial information through a copper-dependent bio-conjugation reaction. Copper-targeted acyl imidazole dyes were designed that operate through copper-mediated activation of acyl imidazole electrophiles for subsequent labeling of proximal proteins at sites of high labile copper to provide a permanent color that resists washing and fixation (Figure 35).

Labile pools have been characterized using this strategy in three main types of brain cells: neurons, astrocytes and microglia. Exposure of each of these cell types to physiologically relevant stimuli distinct changes in these labile copper pools. Neurons exhibit translocation of labile copper from somatic cell bodies to peripheral processes after activation, while astrocytes and microglia exhibit global decreases and increases in intracellular labile copper pools, respectively, after exposure to inflammatory stimuli [108].

This work provides fundamental information on cell-type-dependent copper homeostasis, an essential metal in the brain, as well as a starting point for the design of new activity-transmitted probes for metals and other signal analytes and dynamic stress in biology, in the context of poor regulation of copper in inflammation and neurodegenerative process [109,110,111,112,113]. In Figure 35 the imidazole **1** present in **87** as a metal chelator, in this case copper, is demonstrated through the η1(σ,Npy) interaction, as previously commented.

In a recent background of compounds containing imidazole for anti-proliferative activity, a large number of di/triaryl imidazole-based derivatives (Figure 36) have been reported with p38α MAP and/or BRAF kinase inhibitory activities [114,115,116,117]. Considering these representatives, compound **88** (SB203580) was reported as a p38α MAP kinase inhibitor with IC_50_ value of 48 nM [114,115]. Furthermore, compounds **89** and **90** exhibited their inhibitory activities against p38α at IC_50_ values of 19 and 41 nM, respectively [116]. Compound **91** diaryl imidazole showed inhibitory activity against BRAF (IC_50_ = 900 nM) [117]. However, triaryl imidazole exhibited greater inhibitory activity against BRAF. In a series of triaryl-substituted imidazole-based derivatives, compound **92** was the most potent in inhibiting mutant BRAF in vitro [118].

More studies about compound **92** show a weak inhibitory activity against mutant BRAF at the cellular level. Nevertheless, compound **93** showed potent BRAF inhibition (IC_50_ = 9 nM) and inhibited the growth of BRAF-dependent WM266.4 cells with GI_50_ of 0.22 µM [119]. Takle and coworkers [117] reported on SB-590885 **94** among a series of derivatives based on triaryl imidazole with potent and selective inhibitory activity against BRAF kinase. The compound **94** was prepared to improve water solubility. Furthermore, it showed increased potency with more than 1000 times greater selectivity for p38α, GSK3-β and lck kinases compared to BRAF [117]. Recently, compound **95** (Figure 36) was reported as a derivative based on imidazol-5-yl-pyrimidine with BRAF^V600E^/p38α dual activity [120]. Compound **95** inhibited BRAF^V600E^ and p38α MAP with IC_50_ values of 2.49 and 85 nM, respectively.

The designed compounds **96** and **97** were made by Youssif and collaborators [121] considering the proposed binding interactions of compound **95** with p38α and BRAF^V600E^ (Figure 37). Investigation of compound **88’s** binding interactions in p38α MAP revealed the presence of different types of binding interactions, including hydrogen bonding and hydrophobic interactions [121]. However, an unfavorable donor–donor interaction of the nitrogen *N*-1 on the imidazole ring of compound **88** with Lys53 at p38α was reported. To avoid this unfavorable interaction, the imidazole *N*-1 was alquilated with the hexyl group. Hexyl groups can also provide hydrophobic interactions with hydrophobic residues on p38α. The structure of the compounds **96** and **97** was also extended by a four-atom linker connecting the 1,3-benzodioxolo **96** and 4-methoxybenzyl **97** ring substituents with the triaryl imidazole scaffold [121]. The ligand includes three hydrogen bond acceptor atoms (CO-O-N=) and a hydrogen bond donor group (NH_2_) that can form hydrogen bonds with amino acids on the phosphate biding region at p38α [121].

Among the docking studies, compounds **96** and **97** were able to properly interact with Lys53 in the imidazole scaffold *N*-3. In conclusion, we can see that **96** and **97** are promising lead compounds with high potential for development as new dual p38α/BRAF^V600E^ inhibitors based on the good activity values.

With these recent examples from the literature, we can see that imidazole presents itself as a privileged structure, being widely explored as a nucleus capable of efficiently participating and assisting in the modulation of several molecular targets and acting in other areas, such as probes for biological assessment.

### 3.3. The Origin of Cimetidine as an Imidazole-Containing Drug

To the best of our knowledge, there are four types of histamine receptors that belong to the G-protein-coupled receptor family: histamine H_1_ receptor, histamine H_2_ receptor, histamine H_3_ receptor and histamine H_4_ receptor, which could be stimulated by the endogenous messenger histamine **8** without distinction [122,123]. However, properly designed antagonists must be able to distinguish between them [124]. In the 1960s–1970s, selective antagonists capable of inhibiting the histamine receptors involved in the allergic process (H_1_ receptors) and antagonists capable of inhibiting histamine receptors responsible for gastric acid secretion (H_2_ receptors) were already known [124].

Variations were made in the structure of histamine **8** so that it was recognized by the receptor. Simultaneously, it was able to act as an antagonist. SAR studies on histamine analogs revealed that the requirements for histamine **8** to bind to the proposed H_1_ and H_2_ receptors were slightly different (Figure 38) [125]. On the H_1_ receptors, the essential requirements are as follows:

The side chain must have a positively charged nitrogen atom with at least one hydrogen. Quaternary ammonium salts are not active.

There must be a flexible chain between the cation and the heteroaromatic ring.

The heteroaromatic ring does not have to be imidazole **1** but must have a nitrogen atom with a pair of electrons adjacent to the side chain.

On the other hand, the requirements for H_2_ and H_1_ receptors are the same, except that:

The heteroaromatic ring must have an amidine **5** unit (-HN-CH=N-) acting as a proton transfer agent.

Based on this information, it appears that the terminal amino group being protonated is involved in an ionic interaction with both types of receptors, while the nitrogen atoms of the heteroaromatic ring are linked through hydrogen bonds, and for H_2_ receptors, the ring participates in an extra interaction through proton transfer (Figure 39) [125].

Checking the changes in the structure of histamine **8**, it was noted that 4(5)-methylhistamine **98** is a highly selective H_2_ receptor agonist, showing greater selectivity for H_2_ than for H_1_ [125,126]. 4(5)-Methylhistamine **98**, similar to histamine **8**, is a highly flexible molecule due to its side chain, but structural studies show that some of their conformations are less stable than others (Figure 40) [125,126]. In particular, conformation (**A**) is disadvantaged due to the steric interaction between the methyl group and the side ethylamine chain. The selectivity observed suggests that, for both receptors, 4(5)-methylhistamine **98** has to adopt two different conformations to bind to these receptors [125,126]. As 4(5)-methylhistamine **98** is more active as an agonist of the H_2_ receptor, this suggests that the favorable conformation (**B**) is required for the interaction with the H_2_ receptor, while the conformation required for the H_1_ receptor would be the sterically unfavorable conformation (**A**) [125,126].

In histamine **8**, the exocyclic amino group is protonated at physiological pH, exerting a strong electron-withdrawing effect on the imidazole ring. This effect is more pronounced for the nitrogen closest to this side chain, so the hydrogen atom in the nitrogen *N*π will be more acidic than the one bound to the *N*τ [125,126,127]. For this reason, the last tautomer (*N*τ) is more stable than π (*N*π) in histamine **8**, given that the favored ionization in the structure would lead to stabilization of the charge formed (Figure 41). On the other hand, the thiourea group of burimamide **99** exerts a less pronounced electron-donating effect, and therefore, the *N*π tautomer is favored [125,126,127]. Thus, the idea would be to remove electrons from the burimamide **99** side chain instead of donating electrons through the introduction of an electron withdrawing atom in this chain. The use of an isosteric group of the methylene group was proposed [125,126,127]. The sulfur atom is considered a good classical isostere of the methylene group, as their van der Waals radii and bond angles are similar [125,126,127,128]. The substitution site was also chosen based on synthetic reasons (Figure 41).

The sulfur atom present in the side chain of thiaburimamide **100** carried out the removal of electrons, favoring the *tau* tautomer (τ). Furthermore, a hypothesis has been raised about the possibility of increasing the favoring of this tautomer through the insertion of a group in position 5 of the imidazole **1** ring [125,126,127]. In this position, the inductive effect would have a more pronounced impact on the neighboring nitrogen atom (*N*τ). The methyl group was chosen because it was known that (4)-methylhistamine **98** was highly selective against the H_2_ receptor, in addition to influencing the pKa of hydrogen *N*τ, with an electron donating effect [125,126,127,129]. Metiamide **101** was obtained, showing the highest antagonistic activity on H_2_ receptors (Figure 41).

Metiamide **101** was shown to be 10 times more potent than burimamide **99** [130] and showed great promise as an antiulcer agent. Unfortunately, several patients suffered from kidney problems and granulocytopenia, a disease that leads to a reduction in leukocytes and makes patients more susceptible to infection due to the metabolism of thiourea, which led to metiamide **101** not being approved in stage I clinical trials [131,132].

After the study of metiamide **101**, several other studies were carried out with the objective of modifying the thiourea subunit, a region that is also responsible for important interactions with the target receptor, giving the compounds the profile of antagonists (region of binding of the antagonist). Figure 42 summarizes all the work and rationale used to design the first blockbuster drug cimetidine. Cimetidine, similar to other examples in the literature, is a drug that has an imidazole nucleus in its structure and is a beautiful and inspiring example of rational drug design carried out in the mid-1960s–1970s [133].

Figueiredo and coworkers [134], who described the synthesis of methyl-imidazolyl *N*-acylhydrazone (NAH) derivatives showing antinociceptive activity, demonstrated another interesting study on imidazole **1** tautomerism, exploring the idea of cimetidine **79**. Using the AM1 Hamiltonian [135] with unsubstituted derivative **106**, the conformational behavior of the (NAH) motif around *C*-4 from the imidazole nucleus was investigated. Figueiredo and coworkers observed that the *S*-*cis*-like conformation (**A**) of the *N*-4 tautomer from compound **106** was ca. 6.0 kcal/mol more stable than the corresponding *S*-*trans* (**A′**) conformation (Figure 43) [134]. A five-member-like intramolecular hydrogen bond involving the hydrogen atom of the *N*-acylhydrazone moiety and the nitrogen atom *N*-3 of the imidazole **1** ring could be the reason for this favored conformation. Hydrogen bonding was also observed in the *N*-5 tautomer (**B′**) (Figure 43); however, in this case, a 3 kcal/mol decrease in stability was observed [134].

We observed in this study [134] that for structure (**A**), there are several favorable factors to maintain the (4)-tautomer, i.e., the withdrawing group (EWG) in the *C*-4 position and the methyl group at the *C*-5 position, in addition to the possible intramolecular interaction due to the presence of the NAH subunit. Additionally, the methyl group at the C-5 position may contribute slightly through steric effects to the formation of (**A**) instead (**A′**), given the steric hindrance caused by methyl and the NAH substituent. A similar effect was observed for 4(5)-methylhistamine **98**. Considering all the structures, (**B′**) demonstrates several unfavorable requirements to maintain (5)-tautomerism, even presenting an intramolecular interaction. Given that several of these structural factors play against the pKa value of *N*-1 present in the imidazole ring, its acidity is increased.

## 4. Conclusions

To date, much work has been done on the synthesis, functionalization, description of physicochemical characteristics and biological application of imidazole heterocycles. This work highlighted the special interest and importance of imidazole-containing derivatives in the field of medicinal chemistry and drug discovery.

As demonstrated, imidazole **1** is a structure that, despite being small, presents unique chemical complexity. It is a nucleus that proves to be very practical and versatile in its construction/functionalization and can be considered a rich source of chemical diversity. The role and importance of imidazole **1** in processes for the maintenance of living organisms, such as catalytic participation in enzymatic processes, were also reported. We observed examples of imidazole-based compounds with antibacterial, anti-inflammatory, antidiabetic, antiparasitic, antituberculosis, antifungal, antioxidant, antitumor, antimalarial, anticancer, antidepressant and many other activities in the literature. Finally, the role of imidazole **1** in drug research and development was briefly demonstrated through the discussion of the discovery of cimetidine **79**. It was possible to explore several chemical and biological phenomena of this important drug, which certainly served as a rich base for knowledge and inspiration for several other bioactive imidazole-based drug candidates. This was also demonstrated in the structure of methyl-imidazolyl *N*-acylhydrazone derivative **106**, which was produced after cimetidine and presented a nociceptive effect, and those that continue to currently be developed.

## Figures and Tables

**Figure 1 molecules-28-00838-f001:**
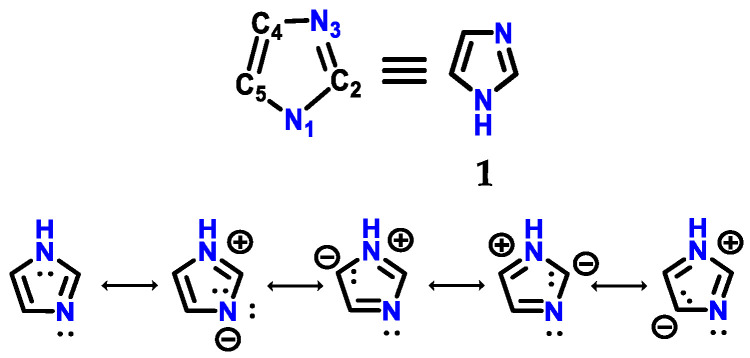
Structure of imidazole **1** with its respective numbering and resonance hybrids.

**Figure 2 molecules-28-00838-f002:**
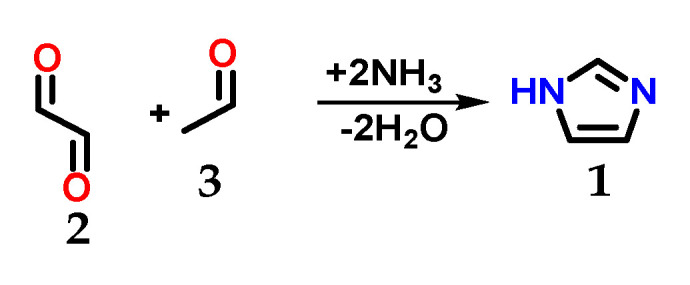
Scheme of the synthesis of imidazole **1** using glyoxal **2** and formaldehyde **3** in ammonia.

**Figure 3 molecules-28-00838-f003:**
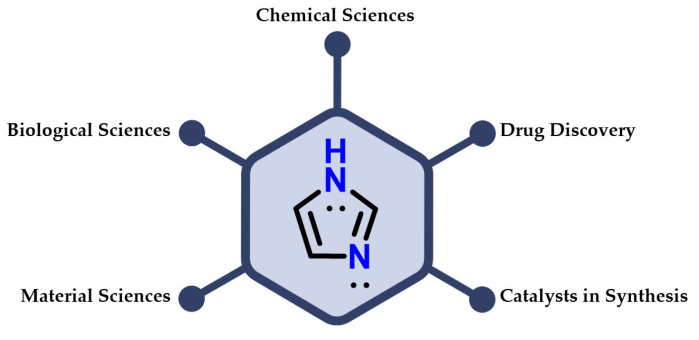
Applications of imidazole **1** in different areas of knowledge.

**Figure 4 molecules-28-00838-f004:**
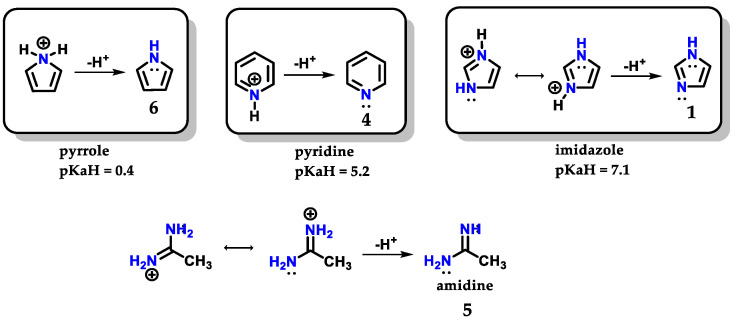
Protonated structures of pyrrole **6**, pyridine **4** and imidazole **1**, with their respective pKaH values and the structure of protonated amidine **5** compared to imidazole.

**Figure 5 molecules-28-00838-f005:**
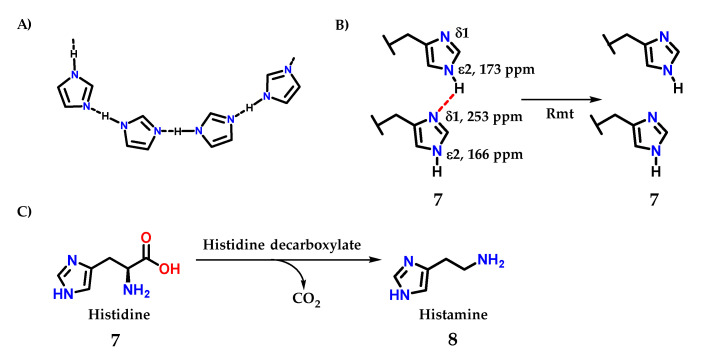
(**A**) Intermolecular interactions through the hydrogen interactions that imidazole-containing derivatives can perform. (**B**) Intermolecular interactions through the hydrogen interactions in histidine residues. (**C**) Structure of histidine **7**, precursor of histamine **8** biosynthesis.

**Figure 6 molecules-28-00838-f006:**
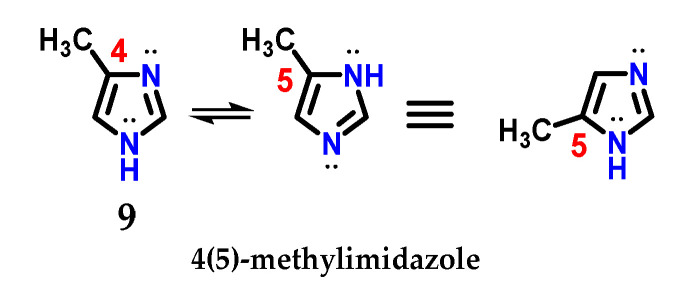
Representation of the tautomeric forms of methyl imidazole **9**.

**Figure 7 molecules-28-00838-f007:**
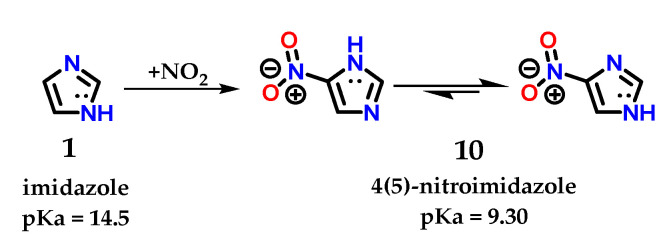
Representation of imidazole **1** and tautomeric forms of 4(5)-nitroimidazole **10**, with their respective pKa values.

**Figure 8 molecules-28-00838-f008:**
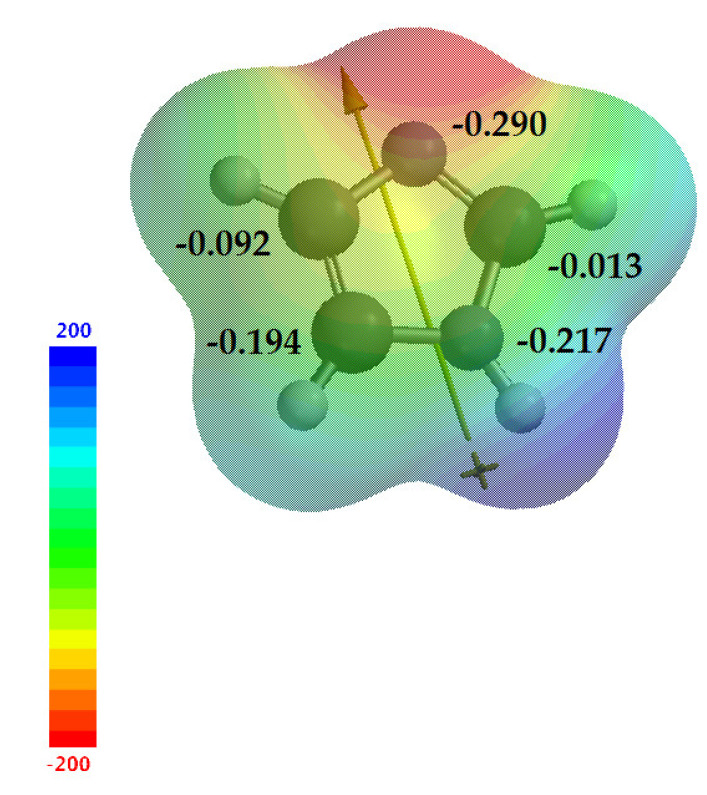
Electrostatic map of imidazole **1**, showing the natural charges of each atom and the dipole moment.

**Figure 9 molecules-28-00838-f009:**
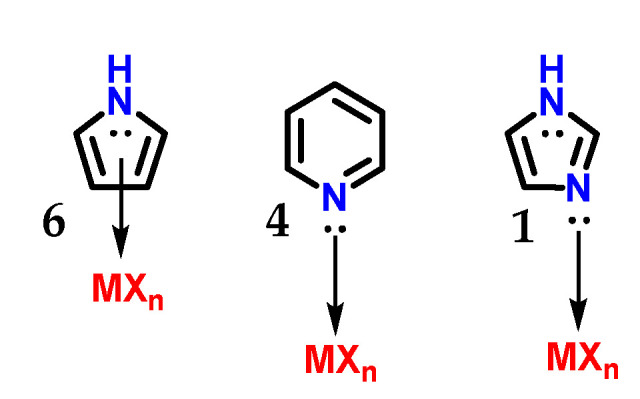
Complex with transition metals η^5^(π) carried out by pyrrole **6** and the complexes η^1^(σ,N_py_) carried out by imidazole **1** and pyridine **4**.

**Figure 10 molecules-28-00838-f010:**
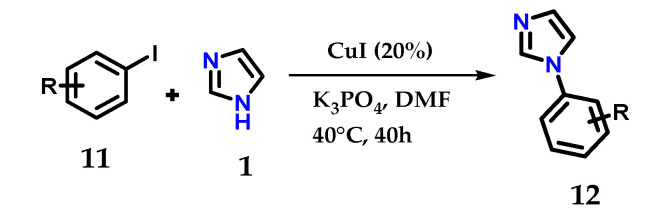
Catalytic *N*-arylation of imidazole **1** with aryl iodides and CuI.

**Figure 11 molecules-28-00838-f011:**
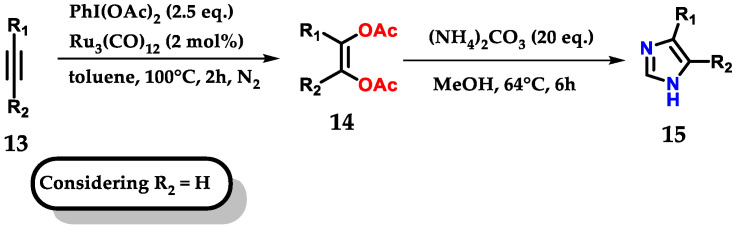
Ruthenium-catalyzed oxidation of alkynes for mono-substituted imidazole **15** synthesis.

**Figure 12 molecules-28-00838-f012:**
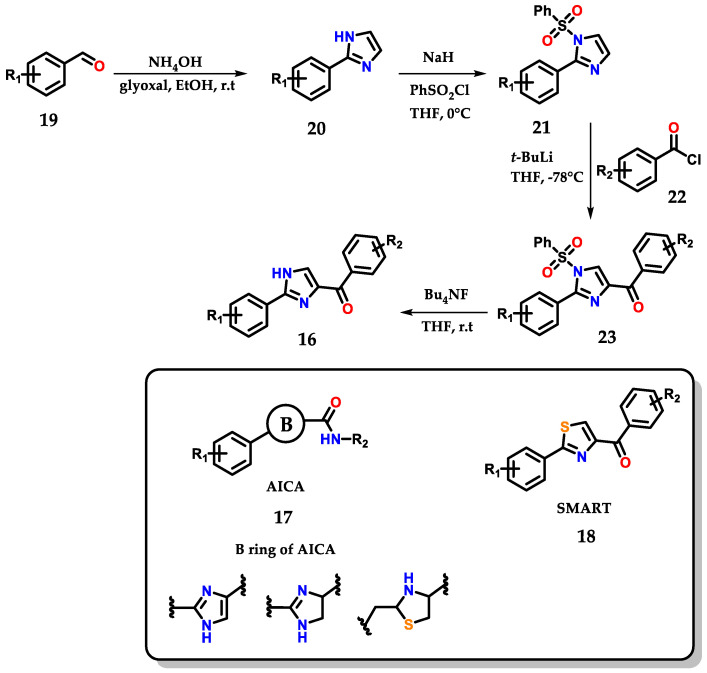
Obtaining 2-aryl-4-benzoyl-imidazoles **16** based on AICA **17** and SMART **18**.

**Figure 13 molecules-28-00838-f013:**
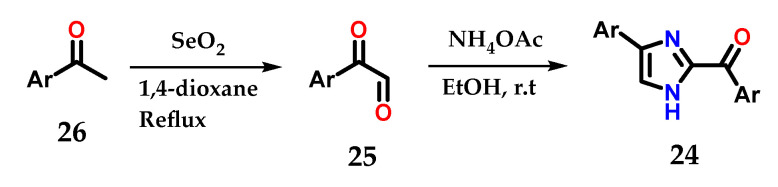
Obtaining disubstituted imidazoles **24** from aryl-methylketones **26**.

**Figure 14 molecules-28-00838-f014:**
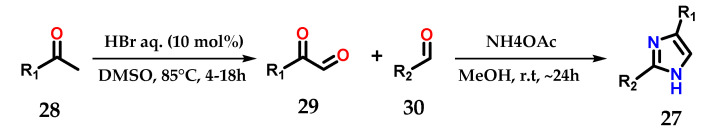
Synthesis of disubstituted imidazoles **27** starting from the acid-catalyzed oxidation of methylketones **28**.

**Figure 15 molecules-28-00838-f015:**
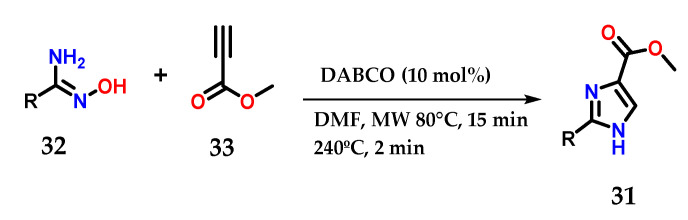
Disubstituted imidazoles **31** were obtained from the base-catalyzed condensation of amidoximes **32** and methyl propiolate **33**.

**Figure 16 molecules-28-00838-f016:**
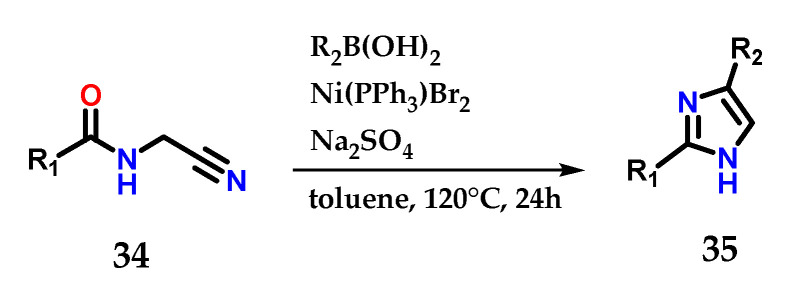
Nickel-catalyzed cyclization of amido-nitriles **34** to obtain disubstituted imidazoles **35**.

**Figure 17 molecules-28-00838-f017:**
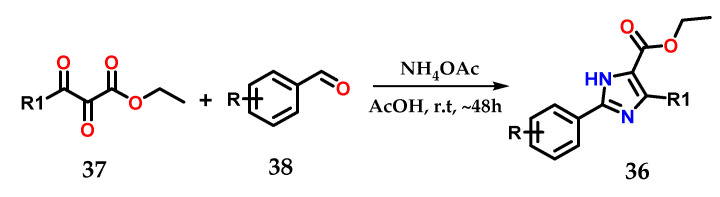
Obtaining 2,4,5-trisubstituted imidazoles **36** from 2,3-dioxo-3-substituted propanoates **37**.

**Figure 18 molecules-28-00838-f018:**
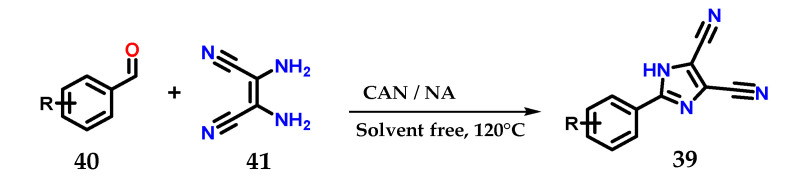
Obtaining 2-aryl-4,5-dicarbonitrile imidazole derivatives **39** using a mixture of CAN:NA as catalysts.

**Figure 19 molecules-28-00838-f019:**
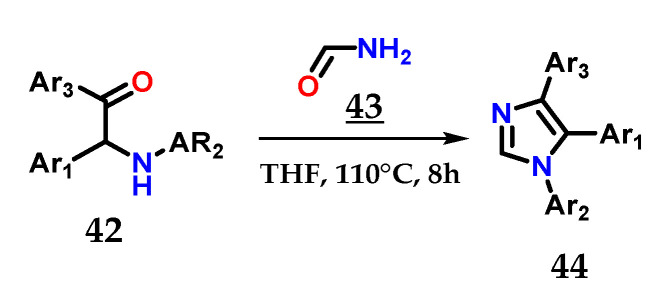
Obtaining 1,4,5-trisubstituted imidazoles **44** from the coupling of α-aminoketones **42** with formamide **43**.

**Figure 20 molecules-28-00838-f020:**
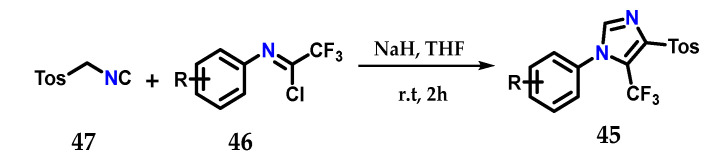
Obtaining 1,4,5-trisubstituted imidazoles **45** using the Van Leusen method.

**Figure 21 molecules-28-00838-f021:**
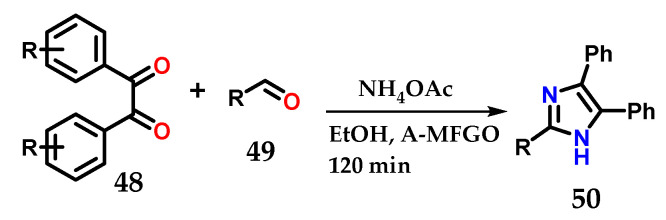
Obtaining 2,4,5-trisubstituted imidazoles **50** from the coupling of substituted 1,2-diphenylethane-1,2-dione (benzyl) **48**, substituted aldehydes **49** and NH_4_OAc using A-MFGO as a catalyst.

**Figure 22 molecules-28-00838-f022:**
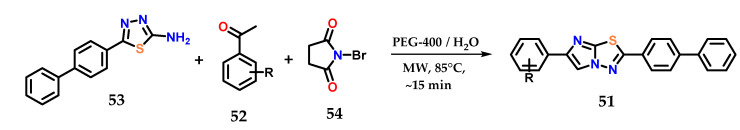
Obtaining benzo[*d*]imidazo[2,1-*b*]thiazoles **51** from the condensation of aromatic ketones **52** and 5-(biphenyl-4-yl)-1,3,4-thiadiazol-2-amine **53**.

**Figure 23 molecules-28-00838-f023:**
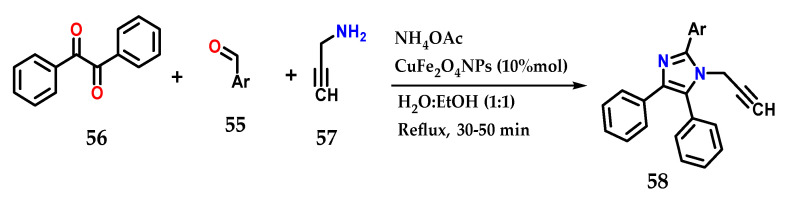
Tetrasubstituted imidazoles **58** were obtained by the coupling of benzyl **56**, functionalized aldehyde **55**, prop-2-ynylamine **57** and ammonium acetate using CuFe_2_O_4_NPs as the catalyst.

**Figure 24 molecules-28-00838-f024:**
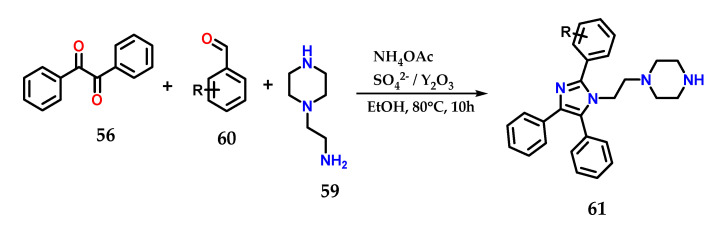
Tetrasubstituted imidazoles **61** were obtained from the multicomponent condensation of benzyl **56**, aminoethylpiperazine **59**, various aldehydes **60** and ammonium acetate using SO_4_^2−^/Y_2_O_3_ as a catalyst.

**Figure 25 molecules-28-00838-f025:**
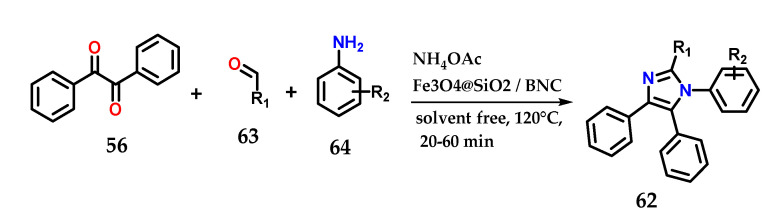
Tetrasubstituted imidazoles **62** were obtained through the condensation of benzyl **56**, aldehydes **63** and anilines **64** in the presence of ammonium acetate using Fe_3_O_4_@SiO_2_/BNC as a catalyst.

**Figure 26 molecules-28-00838-f026:**
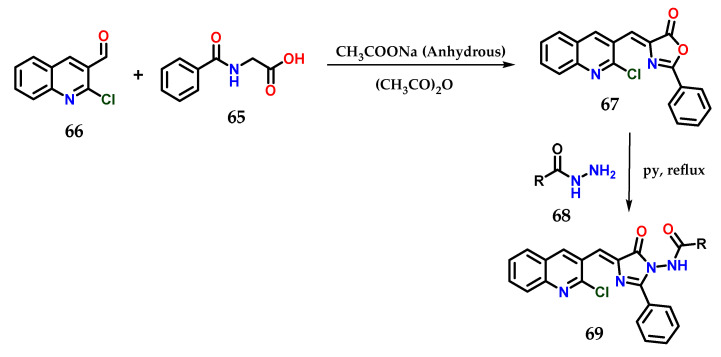
Synthesis of *N*-(4-((2-chloroquinolin-3-yl)methylene)-5-oxo-2-phenyl-4,5-dihydro-1*H*-imidazol-1-yl)(aryl)amides **69**.

**Figure 27 molecules-28-00838-f027:**
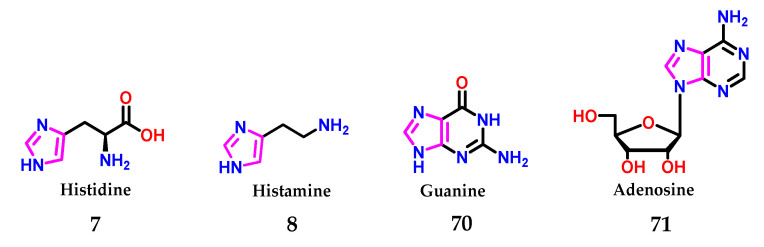
Presence of the imidazole **1** nucleus in several biologically active compounds.

**Figure 28 molecules-28-00838-f028:**
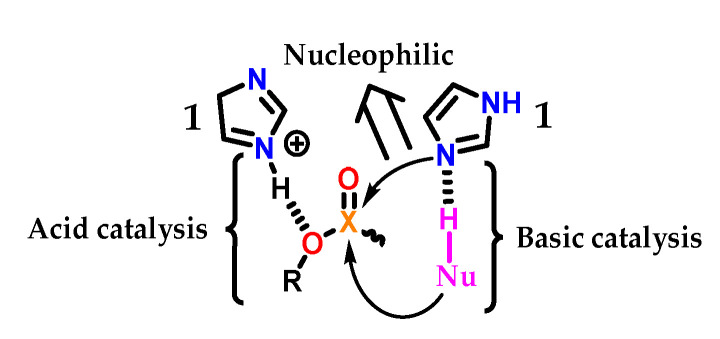
Acidic, basic and nucleophilic catalysis promoted by an imidazole **1** subunit.

**Figure 29 molecules-28-00838-f029:**
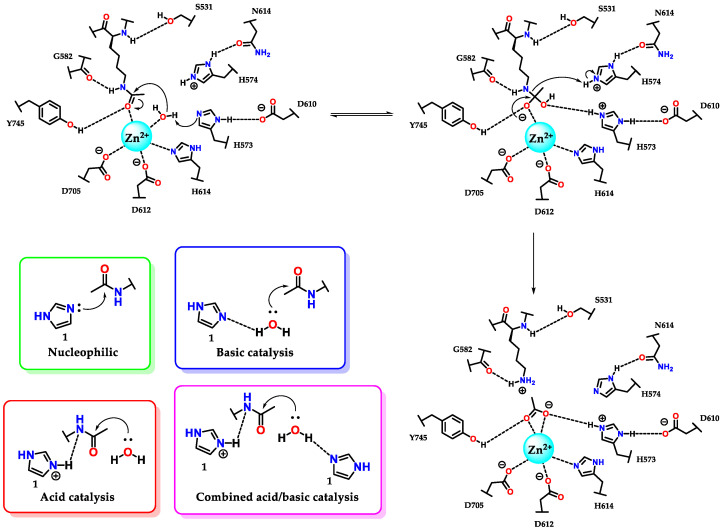
Participation of histidine **7** residues in the deacetylation reaction of lysine residues promoted by enzyme histone deacetylase 6 (HDAC6) and the kinetic profiles of imidazole catalysis.

**Figure 30 molecules-28-00838-f030:**
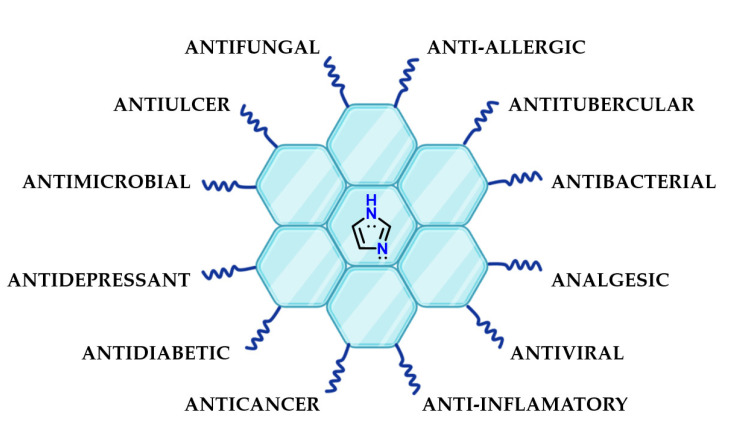
Presence of imidazole **1** in the most diverse classes of bioactive compounds.

**Figure 31 molecules-28-00838-f031:**
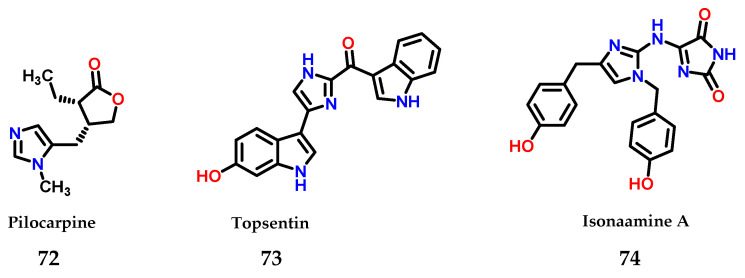
Structures of imidazole-containing natural products pilocarpine **72**, topsentin **73** and isonaamine A **74**.

**Figure 32 molecules-28-00838-f032:**
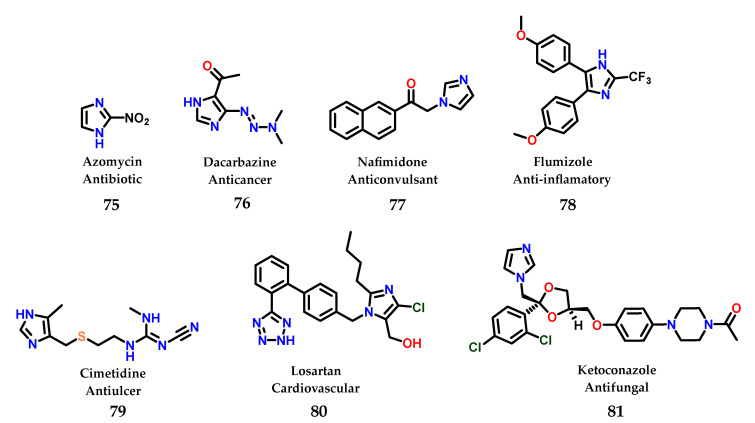
Structures of azomycin **75** and other imidazole-containing drugs currently used in pharmacotherapy.

**Figure 33 molecules-28-00838-f033:**
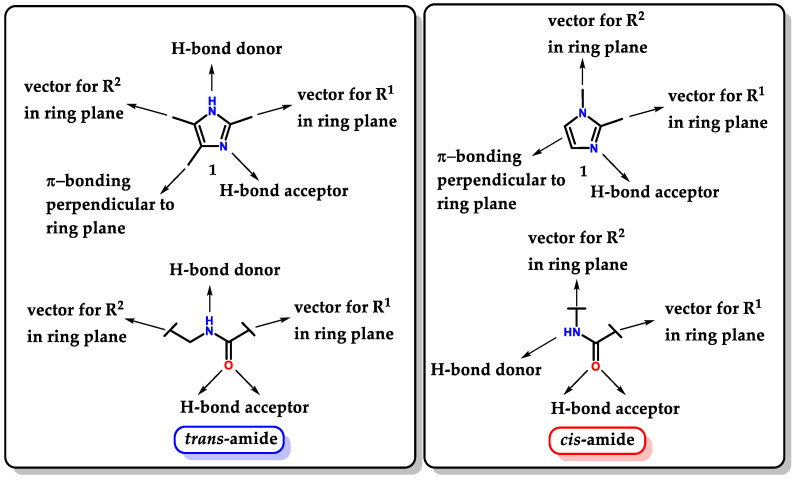
Isosteric relationship between imidazole **1** and *trans/cis*-amides.

**Figure 34 molecules-28-00838-f034:**
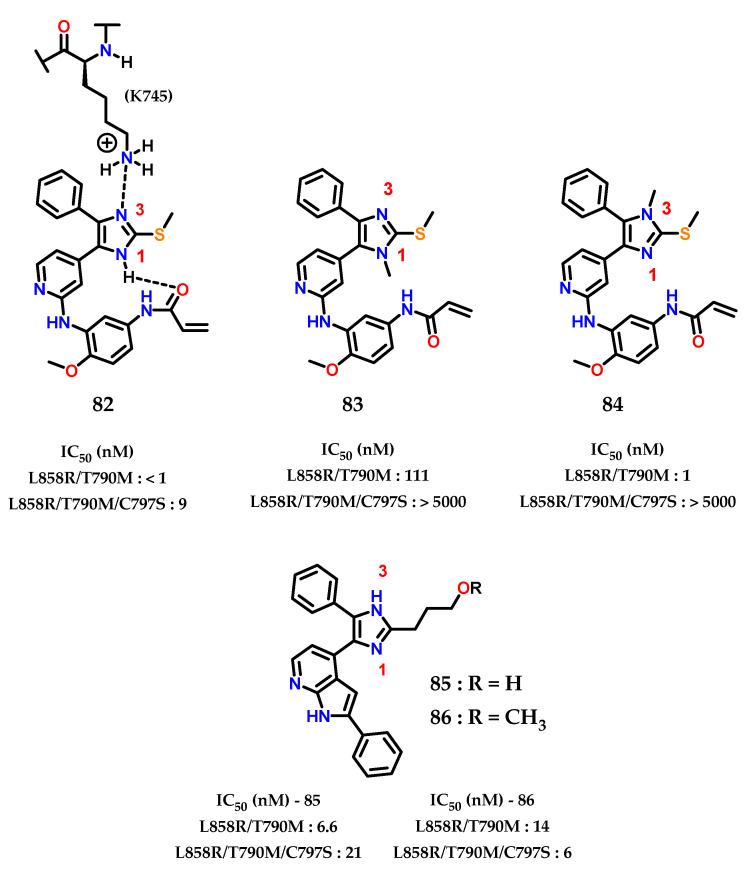
EGFR(L858R/T790M) and EGFR(L858R/T790M/C797S) imidazole inhibitors.

**Figure 35 molecules-28-00838-f035:**
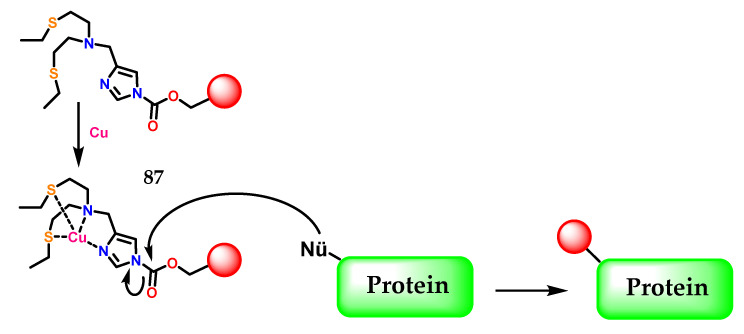
The activity-based sensing (ABS) strategy using acyl imidazole electrophiles **87** for subsequent labeling of proximal proteins at sites of high labile copper.

**Figure 36 molecules-28-00838-f036:**
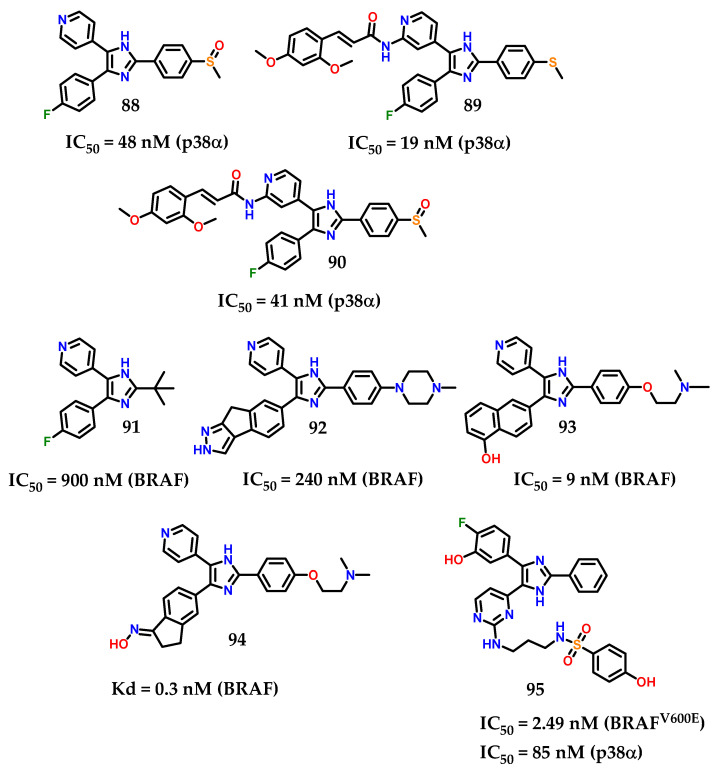
Di/triaryl imidazole-based derivatives with p38α and/or BRAF inhibitory activity.

**Figure 37 molecules-28-00838-f037:**
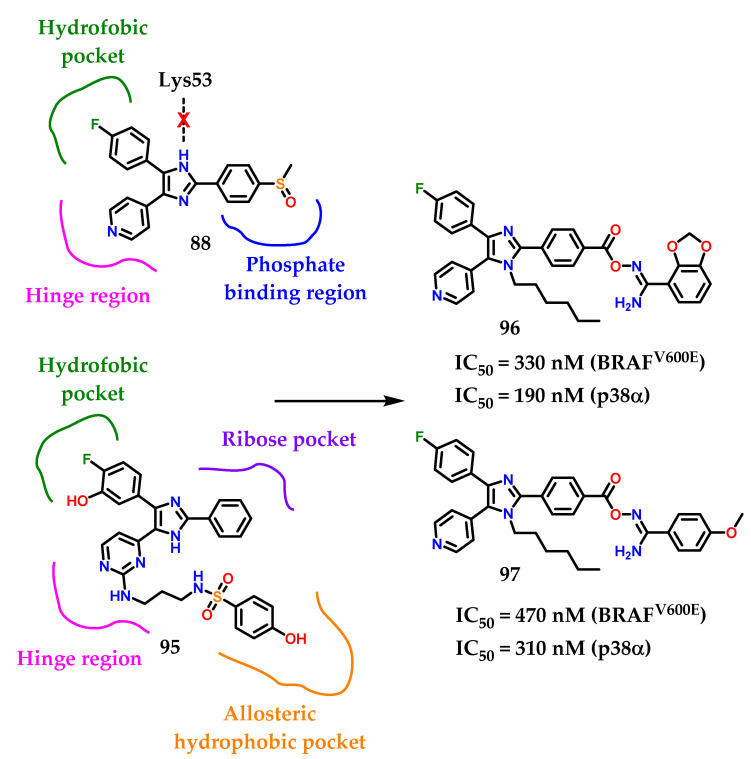
Rational design of compounds **96** and **97**.

**Figure 38 molecules-28-00838-f038:**
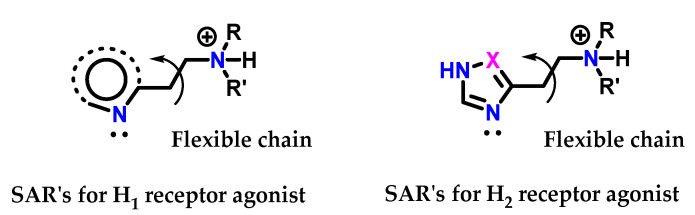
SAR for histamine H_1_ and H_2_ receptor agonists.

**Figure 39 molecules-28-00838-f039:**
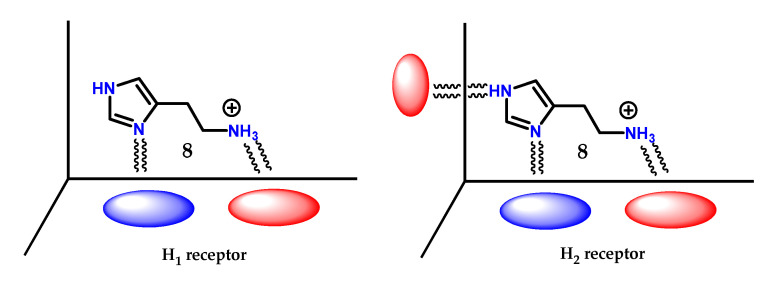
Scheme of the spatial arrangement for the interaction of histamine **8** on the H_1_ and H_2_ receptors.

**Figure 40 molecules-28-00838-f040:**
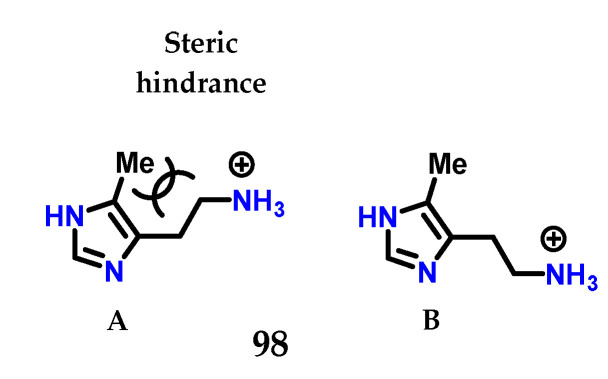
Conformations of 4(5)-methylhistamine **98**. Energetically unfavorable conformation **A** and the more stable conformation **B**.

**Figure 41 molecules-28-00838-f041:**
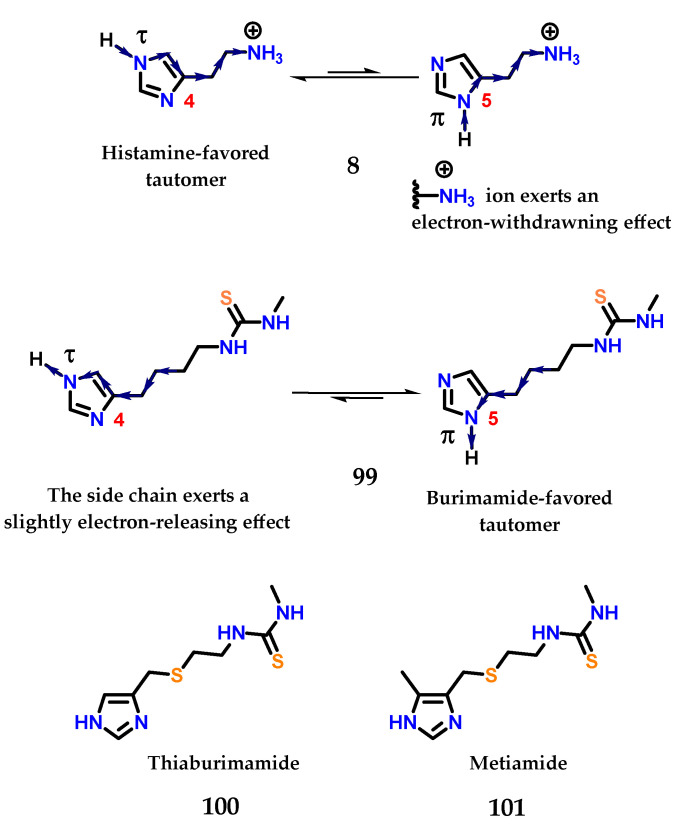
Preferred tautomeric forms of histamine **8** and burimamide **99**. Thiaburimamide **100** and metiamide **101** derivatives as H_2_ receptor antagonists.

**Figure 42 molecules-28-00838-f042:**
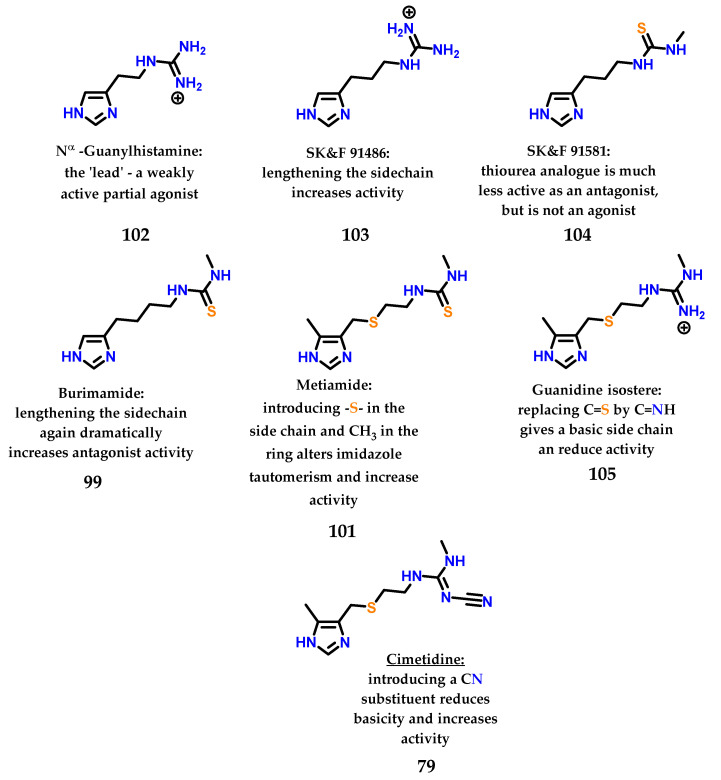
Sequential analogues described by Black, Ganellin and coworkers at SK&F until the discovery of cimetidine **79**.

**Figure 43 molecules-28-00838-f043:**
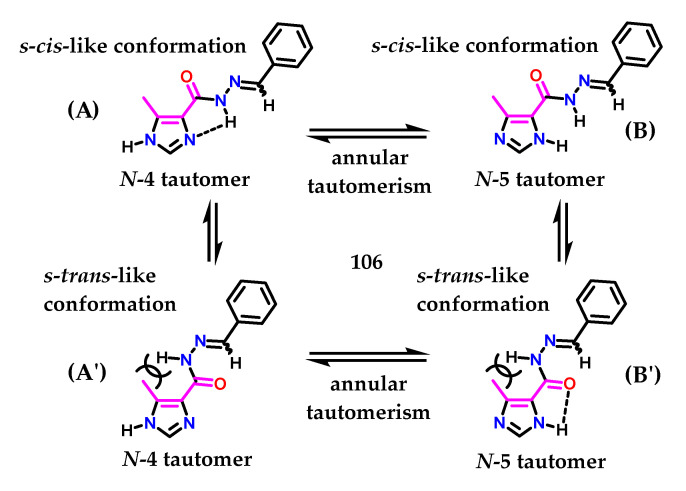
Ring isomerism and conformers of *N*-acylhydrazone derivative **90**.

## Data Availability

Not applicable.

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
