# Peer review of "Imidazole: Synthesis, Functionalization and Physicochemical Properties of a Privileged Structure in Medicinal Chemistry"

_molecules, 2023, doi:10.3390/molecules28020838_

Round 1
Reviewer 1 Report
This paper systematically reviewed imidazole's properties, synthesis methods, and applications in medicinal chemistry and drug discovery. It's a valuable summarization for the future development of imidazole-containing drugs, so I would like to recommend it for publication.
Author Response
There are no questions to answer. Thanks to the reviewer for the comments.
Reviewer 2 Report
This manuscript deals with overview of imidazole chemistry. The contents are similar to that of textbook. This manuscript is helpful for undergraduate students but not for researchers.
As authors said, imidazole is one of the highly important heterocyclic compounds and widely used in various purposes. Indeed, there are numerous papers dealing with imidazole derivatives. In other words, it is quite difficult to cover whole chemistry of imidazole in one review paper. I recommend that authors should focus on special topic of imidazole chemistry.
1) I think ring nitrogen does not use sp3 orbital.
2) When pKa is used for base, the conjugate acid should be placed at the left hand.
3) Totally, reffered literatures are old.
Author Response
Thanks to the reviewer for the comments and suggestions. Regarding the pointed questions:
1) I think ring nitrogen does not use sp3 orbital.
Terms “sp3” have been changed to N-1.
2) When pKa is used for base, the conjugate acid should be placed at the left hand.
Protonated structures in Figure 4, for pKaH data have been shifted to the left.
3) Totally, reffered literatures are old.
Considering a review on the imidazole nucleus, most of the available references are related to the history of the molecule, going through the information already elucidated regarding the physiochemical characteristics as well as the background of the compounds and drugs that have already been developed. The work has about 40 recent references that comprise the time range 2018 ~ 2022. These works report from new synthetic ways to obtain/functionalize imidazole nuclei in the context of medicinal chemistry to new reports in the literature regarding the participation of the imidazole nucleus in active compounds under development.
Reviewer 3 Report
The authors highlighted the imidazole-containing derivatives from their synthesis, functionalization, and physicochemical properties to their application as a privileged moiety in medicinal chemistry. The review is well-organized and would be a good reference resource for those who are interested in imidazole chemistry. Thereby I recommend its publication in Molecules.
Author Response

(The authors gave the same response as above.)

Round 2
Reviewer 2 Report
The manuscript has been improved. So, I think it is acceptable for publication in Molecules.